# Mucosal Immunity of Major Gastrointestinal Nematode Infections in Small Ruminants Can Be Harnessed to Develop New Prevention Strategies

**DOI:** 10.3390/ijms25031409

**Published:** 2024-01-24

**Authors:** P. G. Ashani S. Palkumbura, Thilini A. N. Mahakapuge, R. R. M. K. Kavindra Wijesundera, Viskam Wijewardana, Richard Thiga Kangethe, R. P. V. Jayanthe Rajapakse

**Affiliations:** 1Department of Veterinary Pathobiology, Faculty of Veterinary Medicine and Animal Science, University of Peradeniya, Kandy 20400, Sri Lanka; 2Animal Production and Health Laboratory, Joint FAO/IAEA Centre of Nuclear Techniques in Food and Agriculture, International Atomic Energy Agency, 2444 Seibersdorf, Austria

**Keywords:** helminth, whole parasite, purified parasite products, vaccination, irradiated parasites

## Abstract

Gastrointestinal parasitic nematode (GIN) infections are the cause of severe losses to farmers in countries where small ruminants such as sheep and goat are the mainstay of livestock holdings. There is a need to develop effective and easy-to-administer anti-parasite vaccines in areas where anthelmintic resistance is rapidly rising due to the inefficient use of drugs currently available. In this review, we describe the most prevalent and economically significant group of GIN infections that infect small ruminants and the immune responses that occur in the host during infection with an emphasis on mucosal immunity. Furthermore, we outline the different prevention strategies that exist with a focus on whole and purified native parasite antigens as vaccine candidates and their possible oral–nasal administration as a part of an integrated parasite control toolbox in areas where drug resistance is on the rise.

## 1. Introduction

Gastrointestinal parasitic nematode (GIN) infections are a major constraint in small ruminant production and cause severe economic losses to smallholder farmers. Economically important GIN parasites belong in the order Rhabditida, with a majority found in the family Trichostrongylidae [1,2]. The most economically significant GIN infections in this group are *Haemonchus* spp., *Teladorsagia* spp., *Ostertagia* spp., *Trichostrongylus* spp., *Mecistocirrus* spp., *Nematodirus* spp., and *Cooperia* spp., along with *Bunostomum* spp. and *Oesophagostomum* spp. from the families Ancylostomatidae and Strongylidae respectively (the systematics of nematodes follows Hooda [3]). Most studies on GIN infections in small ruminants have focused on sheep, mainly because goats are more common in developing countries [4,5,6,7,8]. As a result, there is a lack of host-specific information on the species prevalence, geographical distribution, host immune responses, and appropriate prevention strategies of GIN infections in goats [9,10].

*Haemonchus contortus* is a highly pathogenic species in small ruminants in tropical and subtropical regions and can also be found in young cattle and some species of deer [11]. The abomasum is the most affected organ during infections due to the nematodes’ haematophagous nature and the rapid development of worm burdens, making the parasites the most common nematodes that infect small ruminants globally [4,5,6,7,8,12,13]. Haemonchosis begins with the ingestion of third-stage larvae (L3) in pasture. L3 larvae migrate to the abomasum and penetrate glands that line the abomasum before moulting into fourth-stage larvae (L4). Complex immunological responses are elicited at this stage with various factors involved in developing resistance to subsequent infections by the host. These factors are associated with age, breed, and previous exposure to parasites [13]. Young animals are very sensitive to infection when compared to older animals, which acquire immunity after continuous or seasonal exposure [13,14]. In immunized animals, there is a rapid expulsion of parasites before L4 stage larvae are established [15]. *Teladorsagia circumcincta* is a major abomasal parasite species in temperate countries. Immunity can be developed with repeated exposure to *T. circumcincta* and is related to the age of the animal, with weaned lambs highly susceptible to infection [16]. Most animals suppress the infection within 6 weeks after the development of IgE and IgA antibodies [16]. Mucosal mast cells have also been shown to play an important role in preventing larval colonization and the expulsion of adult parasites in association with parasite-specific antibodies [13,17,18]. *Ostertagia ostertagi* is a parasite predominantly found in cattle reared in temperate countries, with a life cycle similar to that of *Haemonchus* sp. The parasite is comparatively smaller, with third-, fourth-, and fifth-stage larvae inhabiting the abomasal gastric glands [19]. *Trichostrongylus* species are a significant problem in small ruminants due to the high fecundity of female parasites, the longevity of larvae in pasture, and a growing resistance to anthelminthic drugs [20]. Infection is initiated with the ingestion of L3 larvae in pasture, which subsequently invade the mucosa, where lesions develop as circular thickened areas several centimetres in diameter [21]. Affected animals are usually asymptomatic but may display diminished appetite and progressive weight loss, with young animals highly susceptible to the infection [22]. *T. axei* is the only parasite of the genus that can be found in the abomasum and has been recorded in cattle, sheep, goats, deer, pigs, and horses. *T. colubriformis* and *T. vitrinus* are primarily found in the intestines of small ruminants [2,23,24]. *Mecistocirrus digitatus* is a blood-feeding trichostrongyle with a pathology like that of *Haemonchus*, although *M. digitatus* adult parasites are much larger at approximately 40 mm and are typically found in tropical climates [25]. These larvae exhibit haematophagic behaviour and heavy infections will cause the common Trichostrongyloidea associated symptoms [20]. *Nematodirus* spp. are primarily intestinal parasites with *N. battus*, *N. fiticolis*, *N. spathiger*, and *N. helvatianus* affecting ruminants. *N. helvatianus* is the only species found in cattle, with the rest infecting small ruminants. These parasites are comparatively less pathogenic, but larval stages usually cause enteritis and intestinal necrosis. *Cooperia* spp. are important parasites in both cattle and small ruminants but are more often found in small ruminants. These are small intestinal worms, and the major species are *C. oncophora*, *C. punctata*, and *C. pectinata*. *Bunostomum trigonocephalum* is a hookworm that occasionally infects sheep and goats. *Bunostomum* infections can occur either via oral ingestion or direct skin penetration by the infective larvae [20]. *Oesophagostomum* spp. are primarily large intestinal nematodes of cattle and small ruminants. *Oesophagostomum columbianum* and *O. venulosum* occur in sheep and goats. Similar to the other GIN infections, transmission can occur via ingestion. Larvae penetrate the intestinal wall and become encysted, forming multifocal nodules throughout the gut, and can remain in the nodules for up to a year. The nodular lesions are used to identify *Oesophagostomum* infections during necropsy [2,20,23]. 

The clinical signs associated with most GIN infections are diarrhoea, anaemia, weight loss, reduced production, growth retardation, oedema, and hyperproteinaemia with severe abomasitis along with the infiltration of mononuclear inflammatory cells, eosinophils into the lamina propria, and hyperplasia of abomasal mucosal cells [2,20,23]. The main control measure applied for all GIN infections is the use of broad-spectrum anthelmintic drugs. There has however been an increase in GIN infections in all ruminants due to emerging drug resistance with a need for alternative methods of disease control [26,27,28,29,30,31]. This review is narrative and discusses publications on the most common GIN parasites with an economic impact on small ruminants. We describe immune responses in the natural host or in experimental surrogates during infection with an emphasis on mucosal immunity. In addition, we cite examples of proven studies on irradiated whole parasites and purified native proteins for vaccination as part of a potential integrated parasite control strategy currently required in regions with increasing parasite drug resistance. This would serve as a solution before more effective methods of control can be discovered. 

## 2. Immune Responses to GIN Infections

Since parasites have evolved within their host for generations, acquiring immunity against GINs is a highly variable and complex process. The development of immunity depends on host-, parasite-, and environment-related factors. Host-related factors include age, immunity, sex, species, and genetic resistance, while parasite-related factors include larval survival, duration of time spent by larvae in the tissues, the release of host immune modulation factors by infecting parasites, and the location of the larvae in the host species. Environmental factors include weather, seasonal climate, and the microbiome environmental conditions in the gut infected with adult parasites [32,33]. The development of immune competence primarily occurs through the prevention of the establishment of newly ingested larvae within the gastrointestinal tract, suppression of parasite growth, expulsion of adult worms, or a combination of all three mechanisms [34,35]. Immunity to intestinal worms also develops more rapidly when compared to abomasal worms [36,37]. It has also been shown that immune responses against GINs in small ruminants is different in sheep when compared to goats [38]. For example, a study that was conducted in sheep in Germany showed GIN infections in farms vary by farm, age, and gender [39]. Other studies in Ethiopia showed a higher susceptibility to *H. contortus* but lower susceptibilities to *T. axei* and *Teladorsagia* spp. in sheep when compared to goats [4,5]. The prevalence of GIN infections was also affected by the agro-climatic zone and breed susceptibility to GIN infections [40,41]. In a different study, seasonal climatic conditions combined with the age of the goats was confirmed to significantly affect the prevalence of GIN infection [42]. Parasitic factors such as the number of parasites used to establish an infection also have a major impact on immunity. A low number of *H. contortus* parasites cause a mild infection when compared to a higher number of parasites with a corresponding escalation of immune responses [43,44]. Furthermore, it has been shown that worm burden is negatively correlated with the number of globule leucocytes in abomasal mucosa during *T. circumcincta* infection in lambs [45,46].

Immune responses can be classified as innate or adaptive. Initial responses belong to the innate immune system, which primarily senses the parasites before the host can initiate an adaptive immune response. The mucosal immune system represents the largest immune organ in the body and defends against more than 90% of infecting pathogens. It is an integrated system and provides a physical barrier to external pathogens with reinforcements from both adaptive and innate immune systems [47,48,49]. Antibodies from humoral immunity and cell-mediated immunity are mainly found in circulation and tissues rather than on mucosal surfaces, with responses mainly carried out by secretory IgA, cytotoxic T cells, and ɣδ cells [47,48]. Innate components of mucosal responses include the mucous barrier, epithelial cell barrier, and the immune cell barrier, which assist in parasite expulsion [48,50]. Adaptive responses cannot provide strong immune protection against infections on mucosal surfaces such as the respiratory system, gastrointestinal tract, and reproductive system [48]. Mucins or mucus glycoproteins are the most important components of the mucus barrier and can be classified as membrane-bound or secreted according to their function and location [51]. Mucous secretions are produced by mucus neck cells in the abomasum and epithelial goblet cells in the small intestine [52,53]. During *Trichinella spiralis* infections in mice, there is goblet cell hyperplasia by the 8th day of infection, leading to an increase in mucus secretion into the small intestine, thus demonstrating increased mucous production against nematode infections [54]. Studies have shown that mucin protects the mucosae from parasite proteolytic enzymes which cause inflammation [49,55]. Phagocytic cells that reside in tissues such as dendritic cells and macrophages play a crucial role in innate immunity and facilitate the initiation of adaptive immunity by sampling the antigens of the parasites at the mucosal surface. Due to the release of chemo-attractants and inflammatory mediators by the innate immune system, mast cells are also recruited to the site of the infection. In addition, chemotactic factors contribute to the recruitment of various inflammatory cells including eosinophils, natural killer (NK) cells, and neutrophils to mucosal sites. In tissues, eosinophils are activated and show directional migration towards parasite infection sites, where they play a role in immune regulation, resistance to parasitic invasion through degranulation, and the release of eosinophil secondary granule proteins that promote the healing of damaged tissue. Eosinophil and mast cell activity is more efficient in secondary infections compared to primary infections [47,56]. The activation of adaptive immune response results in the release of various cytokines leading to T cell proliferation and differentiation. Activation of the adaptive immune system occurs following antigen presentation by antigen-presenting cells such as activated dendritic cells and macrophages [52,57]. T cells play a critical role in the cell-mediated immune responses against nematodes. They are differentiated from lymphocytes by the presence of a T cell receptor on the cell surface. There are several types of T cells, including T-cytotoxic, T-helper, and T-regulatory cells. The principal function of B cells is in the production of specific antibodies against specific antigens. The binding of an antigen to a B cell is followed by the signalling of T helper cells to stimulate lymphocytes to proliferate and differentiate into plasma cells, which produce large amounts of antibodies. Major antibodies that are produced against the nematodes are secretory IgA, IgG, and IgE [58]. 

### 2.1. Involvement of Lymphocytes in Mucosal Immunity against GIN Infections

After exposure to parasitic antigens, antigen-presenting cells such as dendritic cells and macrophages initiate adaptive immune responses that promote the differentiation and proliferation of T lymphocytes. Lymphocytes play an important role in mediating immunity against nematode infections [13]. In a lymphocyte transfer study of sheep, lymph obtained from donors that were hyper immunized with *H. contortus* was transfused into recipients that were subsequently challenged. Recipients who received larger volumes of donor lymph showed a significant decrease in faecal egg output, while those who received smaller donor lymph volumes had significantly higher faecal egg counts [31]. Lymphocyte depletion studies reveal that a reduction of CD4^+^ lymphocyte populations have a marked effect on protection, with reduced mucosal mast cell hyperplasia, tissue eosinophilia, and the production of specific antibodies. A study performed on Gulf Coast native lambs that were treated with mouse anti-ovine CD4^+^ monoclonal antibodies to experimentally reduce lymphocytes showed a higher number of parasite infections when compared to the non-depleted control group [59]. There is a rapid recruitment of CD4^+^ type lymphocytes in the abomasal mucosae during GIN infections [60,61]. This was observed when using sheep for experimental infections with *H. contortus*, which caused extensive hyperplasia in abomasal lymph nodes [47]. A follow up study observed that there was a two-fold rise in the weight of abomasal lymph nodes after infection [62]. Protective Th2 responses are common during helminth parasitic infections [45]. Production of specific cytokines (IL-3, IL-4, IL-5, IL-9, and IL-13), along with the infiltration of eosinophils, basophils, mast cells, and IgE production, are the main features of protective Th2 immune responses to GIN infections [37,63,64]. On the other hand, an increased proliferation of Th1 effector molecules is associated with susceptibility to GIN infection [65,66]. During the course of *H. contortus* infections, resistant sheep produce more Th2-related IL-5 responses, while susceptible animals produce more Th1-related IFN-γ responses [66]. Resistant sheep also display higher densities of mast cells and eosinophils in their mucosa [62]. Comparative studies on local cytokine production during *Teladorsagia circumcincta* infections showed previously exposed animals mounting a Th2 response when compared to naïve animals, which also had a Th2 response but failed to suppress initial Th1 responses, thus making them more susceptible to infection [65]. It has been found that Th2 responses support mucosal immunity and the expulsion of nematode parasites, whereas Th1 responses prolong infections until they become chronic [67]. Shifting Th2 towards Th1 immune responses by using *IL-12* gene transfer was shown to significantly change *T. spiralis* infections in mice by prolonging worm survival and inhibiting muscle hypercontractility along with goblet cell hyperplasia [68]. Additional studies on helminth infection have identified group 2 innate lymphoid cells (ILC2s) as a major component of type Th2 innate responses to infections responsible for IL-5 and IL-3 secretion that induce goblet cell division, increase in mucus secretion, smooth muscle contraction, eosinophil and mast cell recruitment and alternative macrophage activation [37,64,69].

A significant number of activated antigen-specific memory B-lymphocytes will persist after parasitic nematodes have been eliminated from the body. These cells form the basis of immunological memory and can be reactivated much more quickly than naïve lymphocytes and usually provide long-lasting protective immunity against the particular parasite. A lymphocyte depletion study of sheep with *T. circumcincta* showed the sheep remaining immune to subsequent infections due to B cell memory [70]. Changes to the lymphocyte profile during primary infection with L3 worms are much more prominent in the tissues when compared to lymph nodes, whereas during adult infections, a surprisingly small number of lymphocytes are infiltrated into tissue [62]. It was also demonstrated that the activation of CD4^+^ T cells (MHC class II) occurs in the draining lymph nodes of *H. contortus*-infected sheep during a challenge infection with larvae, whereas during *O. ostertagi* infection in calves, antigen-reactive T cell precursor numbers increased dramatically on days 7 to 14 post-infection but decreased to the control level in infected adult animals [62]. WC1 ɣδ^+^ T cells lymphocyte sub-populations are particularly important in non-specific immunity in the abomasal mucosa of ruminants, and their association with protection against *H. contortus* infections has been reported [13]. The common observation in primary infections of most infection models is the increase of ɣδ^+^ T cells, particularly in the tissue when compared to the adjacent lymph nodes [62]. It was shown that there is a reduction in fecundity of female parasites with the increase of WC1 ɣδ^+^ T cells [61]. However, cellular responses in immunized adult animals are more pronounced than in naïve animals during primary infection. These cellular responses manifest as an increase in ɣδ^+^ T cells and CD4^+^ T cells expressing activation markers, with CD8^+^ cells generally remaining stable or decreasing [62]. An increase in T lymphocytes, especially CD4^+^ and ɣδ^+^ lymphocytes, but less in CD8^+^ cells, was observed during *H. contortus* infections in sheep [60]. Increased numbers of CD4^+^, ɣδ^+^ T lymphocytes were found in the abomasal tissue 5 days after infection, with no further increases observed in the adult nematode infected group [1].

### 2.2. Plasma Cells and Antibodies in Mucosal Immunity against GIN Infection

Several antibodies have been shown to correlate with GIN infections, including IgA, IgG, IgM, and IgE, the most important immunoglobulin in the intestine and other mucosal surfaces [58]. Plasma cells in the lamina propria of the intestine synthesize a secretory form of IgA which is transported into the intestinal epithelial cells and secreted directly into the intestinal lumen. IgA forms an immune complex with parasite antigens that have passed the epithelial barrier that are cleared back into the lumen as observed during *H. contortus* infections in sheep [52,67,71]. Additionally, there is an increase in species-specific IgE during nematode infections [72]. A study conducted on sheep with *H. contortus* L3 somatic and excretory/secretory (ES) antigens displayed an elevation of IgE levels after primary inoculation with the ES antigen. Reinfection resulted in a strong increase in ES-specific IgE levels [73]. This was also observed when *Trichostrongylus colubriformis*-specific IgE production is increased in resistant sheep when compared to susceptible sheep [74]. Increased levels of IgA have been positively associated with resistance to *T. circumcincta*, regulating both worm length and fecundity [58]. Resistance is associated with suppressed parasite growth and development mediated by IgA activity against 4th stage larvae. Elevated levels of both IgA and IgG have been observed in *Trichostrongylus colubriformis* challenged sheep. A study performed to determine the levels of parasite-specific IgA in mucosae with resistance to *Haemonchus contortus* found in abomasum reported that they were inversely associated with *H. contortus* worm burden and faecal egg count [71]. IgE antibodies are also commonly associated with helminth infection and are involved in inducing a strong Th2 immune response [75]. IgE-mediated Th1 hypersensitivity responses occur in the gut during chronic infections, which was shown using *T. colubriformis* in sheep infected three times weekly for 9 weeks [76]. Larvae are more effective at inducing antibody responses compared to adult infections. There is an increase in B cells and IgG^+^ plasma cells in the abomasal mucosa at 10 and 13 days after infection with L3 stage *H. contortus* larvae in sheep, which is only seen at day 5 when using adult worms for infection [60]. 

### 2.3. Mast Cells in Mucosal Immunity against GIN Infections

Recruitment and the hyperplasia of the mucosal mast cells are one of the marked features of GIN infection. Mucosal mastocytosis, the presence of intra-epithelial globule leucocytes, suggests that type I immediate hypersensitivity reactions are important in worm expulsion and limiting larval development [77,78,79]. Chemokines and other inflammatory mediators released by the immune cells of the innate immune system cause the recruitment of mast cells to the site of infection. Mast cells are best known for their role in mediating allergic responses. However, an increased number of mast cells has also been observed during nematode infections [71]. A study performed using *H. contortus* in sheep showed that mucosal mast cells usually increase during the course of infection, with mast cell numbers highest when an adult infection is established in the abomasal mucosae [62]. In addition, mast cell responses are higher in secondary compared to primary infections [80]. IL-3 plays a significant role in the development of mast cell responses [81]. Anti-IL-3 antibody treatment in *N. brasiliensis* infection in mice showed decreased mast cell recruitment [82]. Mast cells can respond directly to pathogens and send signals to other tissues to modulate both innate and adaptive immune responses. Two subsets of mast cells have been identified based on their location, connective tissue, and mucosal mast cells. Activation of mast cells occurs predominantly via antigen-induced stimulation of parasite-specific IgE bound to the high-affinity IgE receptor (FcER1) at the mast cell surface [52,77,78,79]. Mast cells produce various types of inflammatory chemical mediators, including histamine, leukotrienes, and proteases. The effects of these chemical mediators are directed toward type 1 hypersensitivity reactions, smooth muscle contraction, increased vascular permeability, local blood flow, and enhanced mucus secretion. In response to nematode infection, mast cells also produce Th2-type cytokines such as IL-13, IL-4, and IL-5, which contribute to the recruitment of inflammatory cells such as eosinophils, natural killer (NK) cells, and neutrophils to the mucosal sites [64]. 

### 2.4. Eosinophils in Mucosal Immunity against GIN Infection

Unlike mast cells, eosinophils usually maintain a consistent population of cells, with mature eosinophils continuously recruited at a low level into the blood from the bone marrow to the target tissue. During GIN infections, there is a dramatic increase in the number of eosinophils which migrate to the parasitic targets [83]. There is a pronounced eosinophil response after stimulation by GIN infections such as *H. contortus* infection in sheep. Interleukin-5 (IL-5) is the main cytokine responsible for the dramatic, T cell-dependent increase in eosinophils (eosinophilia) in blood and tissues during *H. contortus* infections in sheep [71,84]. The major function of eosinophils during nematode infection is a direct cytotoxic effect on the parasite, particularly through the release of the granule proteins that mediate vasodilation, smooth muscle contraction, and mucus secretion. In addition, eosinophils play a minor role in tissue remodelling and regulation [71,84]. *H. contortus* infections in sheep showed that eosinophil recruitment into abomasal mucosae is more prominent in the early phase of the disease where larval stage development occurs when compared to adult worm infections [62]. Eosinophil responses have also been shown to be more pronounced in secondary infections during *Ostertagia* studies in sheep [56].

### 2.5. Mucosa-Associated Lymphoid Tissue (MALT) in the Development of Immune Responses against GIN Infection

Half the lymphocytes of the immune system are found within mucosa-associated lymphoid tissue (MALT). MALT is present on the surfaces of all mucosal tissues. Its most common examples are gut-associated lymphoid tissue (GALT), nasopharynx-associated lymphoid tissue (NALT), conjunctiva-associated lymphoid tissue (CALT), and salivary duct-associated lymphoid tissue (DALT) [85]. MALT is functionally divided into two categories, effector sites and inductive sites. GALT, BALT and NALT, CALT, and DALT are considered inductive sites. Inductive sites have secondary lymphoid tissues where IgA class switching, and clonal expansion of B cells occur in response to antigen-specific T cell activation. After activation and IgA class switching, T and B lymphocytes migrate to effector sites. Effector sites can be found in all mucosal tissues as disseminated lymphoid tissue diffusely distributed throughout the lamina propria [85,86]. Peyer’s patches and lymph glandular complexes are the primary inductive sites in the intestinal tract [87,88]. Peyer’s patches are randomly distributed throughout the mucosal and submucosal layers of the intestinal tract, but the highest density is found within the jejunum and distributed along the anti-mesenteric border of the jejunum [87,88]. A study conducted using healthy goats found that the number of CD4^+^ T cell numbers is higher in the areas near ileal Peyer’s patches and mesenteric lymph nodes [89]. Isolated lymphoid follicles are located in the anti-mesenteric border of the small intestine with crypto patches; lymphoid aggregates found in the inter-cryptal lamina propria of the small intestine with T cells and dendritic cells [88]. Lympho-glandular complexes in the colon resemble Peyer’s patches. However, they are smaller and have fewer follicles with smaller germinal centres. Crypts extending into the colonic submucosa lined with follicle-associated epithelium and surrounded by lymphoid tissue may occasionally be evident in the distal colon and are randomly distributed with an average of 1.4 patches per centimetre of the colon [88].

### 2.6. Cytokine Response against the GIN Infection

The majority of intestinal nematodes elicit a Th2 immune response. Type 2/Th2 immunity is characterized by the specific cytokine response along with the release of interleukins. There are two subtypes of T helper cells, Th1 and Th2, based on their cytokine production. Th1 cells produce gamma interferon, IL-2, and alpha lymphotoxin. Th2 cells produce IL-4, IL-5, IL-6, IL-9, IL-10, and IL-13, where cytokines produced by Th2 cells correlate negatively with those produced by Th1 cells and vice versa [90,91,92,93]. During a nematode infection, the predominant immune response is mediated by Th2 cytokines. The main cytokines that are involved in the Th2 immune response are IL-4, IL-5, IL-9, and IL-13. Other than the adaptive immune system, innate immune cells also play a role in mediating cytokine production. Mast cells also produce type 2 cytokines such as IL-4, IL-5, and IL-13. Basophils are also a major contributor to IL-4 secretion, and innate lymphoid cells are also a source of IL-5 and IL-13 [83,90,93]. IL-4 and IL-13 will increase the sensitivity of the target cells to mast cells and basophil-derived chemical mediators. In addition, IL-4 and IL-13 promote the increased contraction of the intestinal smooth muscle cells, increased permeability of the intestinal epithelial cells, and goblet cell hyperplasia. When IL-4 is present in the extravascular spaces, it induces activation of tissue macrophages and aids in tissue repair and wound healing [76,77]. IL-5 stimulates the increased production of eosinophils and triggers the secretion of IgA from the B lymphocytes. IL-13 increases the repair of the damaged epithelial tissues and the production of mucous [52,91]. IL-9 causes an increased level of recruitment of the mucosal mast cells [52,91]. A study that was performed using *Trichostrongylus colubriformis* determined the gene expression of IL-4, IL-5, IL-10, IL-13, TNF-α, and IFN-γ in the intestinal lymph of sheep using qPCR [94]. Resistant sheep had consistently higher IL-13 production which changed significantly between the first and second challenge infections. IL-13 and IL-5 genes were strongly up-regulated in animals with previous exposure to the infection. Genes for TNF-α and IFN-γ also showed mild up-regulation [94]. A study performed using *H. contortus* in lambs confirmed that there was an obvious rise in IL-4 and IL-13 levels in the abomasal mucosae following the infection [95].

### 2.7. Detection and Recognition of Nematode Parasites

Epithelial cells are the first cell type in the host that come into contact with parasite larvae that break through the mucous barrier. The ability of infected cells to respond to nematode parasites is still not well understood. Healthy epithelial cells that are located adjacent to the diseased cells might sense the parasite-derived secretory products/molecules and tissue originated and damage associated molecules, leading to the initiation of the inflammatory cascade [77]. It has been shown that specialized chemosensory epithelial cells called Tuft cells proliferate during infection, which is critical for transmitting the initial signals to develop type 2 immunity within the host [77]. Germ line-encoded pattern recognition receptors (PRRs) are crucial in the innate immune system for the detection of the parasite. The best examples of pattern recognition receptors are C-type lectin receptors (CLRs) and toll-like receptors (TLRs). CLRs and TLRs are found in many cell types, such as the cells of mucosal surfaces, and immune cells, such as antigen-presenting cells (APCs), macrophages, and dendritic cells [96]. PRR proteins detect both pathogen-associated molecular patterns (PAMPs) and damage-associated molecular patterns (DAMPs). Both PAMPs and DAMPs can result in the induction and the continuation of inflammatory processes. In helminths, molecules such as lipids and glycans serve as PAMPs [97,98]. PRRs, molecules that are responsible for initially identifying the PAMPs of parasites, are also important in the initiation of cytokine release and induction of other signals which are responsible for the activation and manipulation of the adaptive immune system. In *H. contortus*, three fucose residues associated with the core region of N-glycan residues of H-gal-GP act as parasite pattern recognition residues that induce both innate and adaptive responses after recognition and phagocytosis by macrophages [99,100]. The type of adaptive immunity developed against the infection depends on the initial interaction between the cells of the innate immune system and parasitic antigen. Specifically, dendritic cells are responsible for presenting parasite antigens for T lymphocytes [101]. More studies into the role of PRRs in the response to nematode infection are required.

### 2.8. Expulsion and Removal of Nematode Parasites 

After the establishment of adaptive immunity within local lymph nodes around the intestinal tract, activated effector cells migrate into the infected area to carry out the expulsion of the nematodes. This mechanism of ejection is a multifaceted process, which involves increased production and secretion of mucous by goblet cells, the release of neutralizing chemicals by white blood cells and epithelial cells, hyper-proliferation of the epithelial cells, and an increase in the peristaltic movements of the intestines [77,102]. The process is basically initiated by IL-13 and IL-4 [103]. Mucous or mucins trap the parasites by blocking their motility, although some parasites can still penetrate the mucous layer, proving that parasites have evolved strategies to escape this barrier. In *T. muris* infections, the parasite is still able to penetrate the mucous layer with the secretion of serine proteases that are capable of digesting mucin 2 in the mucous layer [104]. Increased production of mucous is mostly driven by IL-13, IL-17, IL-4, and IL-22 [77,102]. In addition, IL-13 and IL-4 increase smooth muscle contractility in the gut during nematode infections [91]. This was confirmed in mice infected with *Heligmosomoides polygyrus* larvae, where IL-4 and IL-13 caused an increase in smooth muscle contractility in the small intestine [105]. The release of various protein chemicals and specific antibodies by activated white blood cells is also important. These products elicit toxic effects on the parasite and aid in its expulsion [77,102]. After being entrapped by the mucous layer and exposed to various chemical products/proteases and neutralizing antibodies, parasites are expelled from the gut via increased intestinal peristaltic movements [77,102]. In a *Haemonchus contortus* infection, the expulsion process can be categorized into two major immune-mediated mechanisms, rapid rejection, and delayed rejection [13]. Rapid rejection occurs when there is a well-established form of immunity in the host. When the host is hyper-sensitized via repeated larval (L3) infections for a prolonged period, rapid rejection of subsequent larval (L3) infections results within 48 h [13]. This type of reaction is mediated by the type I hypersensitivity reaction with the involvement of IgE antibodies, mucosal mast cells, and goblet cells and increases the peristaltic movements in the gut, preventing the establishment of new larvae in the crypts of the abomasum [13,106]. Delayed rejection is characterized by the recruitment of the eosinophils direct eosinophil-mediated larval killing through antibody-dependent cell cytotoxicity (ADCC) [13]. Delayed rejection takes place when the ingested larvae (L3) penetrate immune barriers and elicit a primary infection within the host or when a sensitized host is unable to acquire the required level of immunity against the infection [13]. In this process, activation of local lymph nodes and recruitment and activation of lymphocytes (CD4, CD25, B cells) and eosinophils occur [13]. Once the nematode infection is cleared from the gut and the inflammatory process is resolved, reparative mechanisms are initiated for the restoration of the damaged tissues. This process is mainly driven by type 2 cytokines, macrophages, and eosinophils [102]. In a study that was conducted in mice on muscle regeneration after inducing muscle damage with cardio-toxins, the rapid recruitment of eosinophils, secretion of th2 cytokines (IL-4 and IL-13), macrophage activation in the reparative process was observed [107].

## 3. Major GIN Infection Control Strategies against Nematode Infections

Control measures used against GIN infections are categorized as chemical, biological, and vaccination methods [108]. A fourth category that combines elements of these three is referred to as integrated parasite control. Chemical control is carried out using anthelmintic drugs and is the common method used to control GIN infections globally. Anthelmintics can be applied either as chemotherapy in infected ruminants or for chemoprophylaxis, a pre-emptive measure in susceptible animals against potential parasitic infections. However, the continued usage of the same anthelmintic drug has caused the rapid development of anthelmintic resistance by the nematode parasites, food safety issues due to drug residues, and a limited availability of the effective drugs due to their high cost [108]. Other studies have administered drugs as topical applications to control nematode infections [109]. A study conducted in sheep with eprinomectin showed that it was effective against a host of nematodes, including *H. contortus*, *N. battus*, *N. spathiger*, *O. venulosum*, *T. circumcincta*, *D. filarial*, *C. curticei*, *T. axei*, *T. colubriformis*, and *S. papillosus* [87]. The use of eprinomectin orally in goats has been common due to their high susceptibility to GIN infections [30]. It has been argued that doses administered are suboptimal, with indications that this may lead to nematode resistance, which is more likely to emerge in goats when compared to other ruminants [110]. As an alternative, natural plant-based agents have been used as an alternative for controlling nematodes [111]. Herbal formulations of leaves from *Azadirachta indica* and *Nicotiana tabacum*, flowers from *Calotropis procera*, and seeds from *Tachyspermum ammi* have been shown to reduce the percentage of nematode eggs by 42%, 52%, and 70% respectively [112]. Tanniferous plants or tannins have also been used as an alternative approach for controlling the infection with varying degrees of success [113,114]. Other biological prevention measures include the introduction of the nematophagous fungi (*Duddingtonia* spp.) into pasture to reduce the pre-parasitic stages of GIN infections [108,113,114]. Integrated parasitic control measures utilise chemical, biological, and biotechnological control methods to reduce the extensive usage of chemicals and to achieve long-lasting protection in susceptible animals. A common example is the combination of both anthelmintic treatments and grazing management practices which together facilitate a more effective way of controlling parasites when compared to using them alone [108].

### 3.1. Vaccination as a Parasite Control Method

Vaccination could be considered the safest and most cost-efficient way to control GIN infections, as it offers a natural and chemical-free method that does not contaminate grazing pasture and is an effective prophylactic method when compared to anthelmintics, which can lead to parasite resistance with repeated use [29,108,115]. Vaccination using the irradiated bovine lung worm *D. vivaparus* has been successful when compared to previous attempts that used antigen preparations from adult worms, thus indicating the importance of hidden and conformational epitopes that can only be presented by living but non-infectious parasites produced via irradiation [116]. With the commercial success of irradiated *D. vivaparus* as a vaccine, other irradiated GIN trials have been attempted, including gamma-irradiated *B. trigonocephalum*, *Oe. Columbianum*, *T. colubriformis*, and *H. contortus* in sheep; *A. caninum* in dogs; and *Trichinella spiralis* in pigs, with varying levels of success as summarised in Table 1. Nematodes are large extracellular parasites with complex genomes that are required for escaping various immune responses for successful establishment in the host [37]. An effective vaccine would need to stimulate many different host immune pathways to sufficiently survive subsequent infection. Using whole irradiated L3 larvae that have lost their ability to establish an infection but are still alive and therefore mimicking a natural infection, ensures that the host is able to induce all the variable, but pertinent immune response is required to prevent infection [117]. It is therefore not surprising that most of the experimental vaccines listed on Table 1 do not confer sterilising immunity but rather reduce parasite shedding to a minimum which, when combined with other integrated control measures, would effectively stop all infection in a herd; e.g., irradiated *Trichostrongylus*, with an efficacy of almost 80%, is sufficient to prevent disease [118]. In order to reduce shedding, three parameters need to be fulfilled. First, the larvae need to be less infective with a compromised ability to establish as adults. In addition, the number of worms shed needs to be lower compared to natural infections due to compromised pathogenicity after irradiation, and lastly, reduced fecundity amongst adult female worms that results in decreased egg numbers must occur [119]. When *A. caninum* L3 larvae irradiated at 400 Gy were used as a vaccine (Table 1), approximately 75% of the irradiated larvae failed at gut establishment after dying in the lungs, with larvae in vaccinated dogs a great deal less motile and thus unable to evade the vaccinated host immune response when compared to naïve dogs [120,121,122,123,124]. The presence of dead larvae in the lungs of vaccinated dogs stimulates a strong immune response, and although they continue to shed parasite eggs when challenged, the numbers are much lower when compared to a natural infection, and they do not display any of the symptoms seen during natural infection [123]. In the case of *H. contortus*, two vaccinations of 10,000 parasites irradiated at 600 Gy resulted in protection of up to 86% in sheep older than 6 months (Table 1). This was however not replicated in 2-month-old lambs, with poor responses upon challenge [125,126]. Further experiments with lower radiation doses were also explored [127]. Revived studies using *H. contortus* parasites irradiated at the lower dose of 200 Gy was effective when used to immunise 4-month-old goats [117]. This would suggest that previous experiments that did not test the viability of irradiated parasites before inoculation failed to provide younger animals with crucial conformational antigens that are necessary for inducing protection. Using doses that kill instead of producing live but non-infective parasites has negative consequences, especially when immunising younger animals [117]. It would therefore be advisable to analyse the viability of irradiated parasites before immunisation when producing commercial batches. Low dose exposure at 100 Gy of 50 thousand *T. spiralis* parasites that were used to immunise pigs prevented infection when challenged (Table 1). A study comparing murine *Trichinellosis* vaccination with parasites irradiated at 300 Gy to treatment using *Punica granatum* confirmed how effective using irradiated parasites can be in preventing infection [128]. Vaccinated mice exhibited a significant reduction in muscle larvae at 72.5% when compared to a treated group at 56.3%. A combination of vaccination with treatment as an adjuvant was postulated to give higher levels of protection [128]. A dose-dependent response was observed when using *T. colubriformis* parasites irradiated at 600 Gy (Table 1). A threshold of more than 5000 parasites is required to generate immunity, with success seen at 2 doses of 20,000 parasites [129]. A study using gerbils as a model host for ruminant GIN infections revealed strong mucosal antibody responses following vaccination and challenge [118]. 

The ruminant host immune system can also be stimulated by ES parasite antigens/proteins that can be utilised as successful vaccine candidates compared to using whole parasite vaccines. In the case of *H. contortus*, the most successful of these has been using soluble adult parasite gut antigens, commercially available as Baebervax^®^ (Table 2) [13,142,143,144,145,146,147,148,149,150]. Vaccines based on purified or recombinant proteins have also been developed for various GIN infections, ranging from *T. circumcincta*, *T. colubriformis* to *Ostertagia ostertagi* with varying degrees of success as shown in Table 2. In some cases, such as *O. ostertagi*, the recombinant version of the purified protein did not elicit the expected response and performed subpar when compared to the purified native protein, indicating the complexities involved in developing parasite vaccines [151]. Due to their complex structure and pathogen–host interactions, GIN vaccines require various considerations. This includes understanding the exact level of protective immunity that should be induced by the vaccination, selection of the most suitable antigen able to elicit an effective level of immunity, and selection of the most appropriate vaccine formulation, such as adjuvants, to obtain optimal success in a combination of the antigen and mode of delivery [108,152]. The development of immunity to the appropriate vaccine antigen will however not perform equally in all ruminant recipients, as this will further depend on various host factors. These include age, breed, genetics, individual animal variation, the plane of nutrition, health status, level of management, climate, and the complexity of the parasite vaccinated against [108,147,152,153]. Native parasite antigens as vaccine candidates and commercially viable subunit vaccines for ruminant GIN infections have recently also been discussed in recent publications [154,155,156].

### 3.2. Mucosal Immune Responses in Vaccine Development

The major effector mechanisms, especially in live vaccines, are the production of antibodies, induction of strong CD8^+^ T cell responses, and CD4^+^ T cell responses against the infection. However, in most situations, inactivated or killed vaccines do not produce a sufficient quantity of antibody titres in mucosal surfaces [165]. Exposure of GINs to mucosal surfaces often facilitates the development of mucosal immunity in both the innate and adaptive arms of the immune system [166]. However, almost all vaccines developed to date have been applied to the host through non-mucosal parenteral routes, directly to the cells of the immune system in tissue or blood where the major effector mechanism of mucosal immunity is not secretory IgA, but CD4^+^ and CD8^+^ T lymphocyte-mediated immunity. More than 80% of the entire mucosal lymphoid population is comprised of T lymphocytes, while CD4^+^ Th2-type responses are more effective in inducing mucosal immunity in sheep. In addition, eosinophil production, mast cells primed with specific IgE production, increase in the smooth muscle contractility, increase in the epithelial cell turnover rate, and mucus secretion are other effector mechanisms involved in parasite expulsion [52,64,166]. For gastrointestinal tract diseases, particularly parasitic infections, the oral route will be the preferred route of vaccination, as this would mimic how GIN infections are acquired and this would generate an effective immune response on mucosal surfaces. Intranasal vaccines are used to induce a rapid interferon response basically due to the higher concentration of NALT in the respiratory lymphoid system. The absence of interference with maternal antibodies will be an added advantage of intranasal vaccines. Induction of NALT also has the effect of inducing immunity in other mucosal surfaces because of the common mucosal immune response theory [167,168]. Oral vaccines are exposed to NALT as a major portal entry, where they induce a rapid immune response. Immunization of GALT is largely dependent on the reaching time of the vaccine to the lymphoid tissue such as Peyer’s patches. If oral vaccines were administered within the first 24 h of birth, there would be a high risk of neutralizing the vaccine by the maternal antibodies of the colostrum [48]. Several experimental vaccines have attempted mucosal application. In a study conducted with *H. contortus*, mucosal antigen extracted from the parasite was inoculated into the abomasum and rectum via intra-mucosal injections; significant lymphocyte proliferation in the abomasal mucosae was detected following vaccination [169]. In another study, 3-month-old lambs were immunized via the intranasal route with a recombinant part of the catalytic region of the serine/threonine phosphatase 2A (PP2Ar). The immunized lambs showed a strong immune response with reduced faecal egg count against *Haemonchus contortus* and *Teladorsagia circumcincta* parasites, establishing the possibility of using the intranasal route to induce immunity against GIN infections [170]. Other additional advantages of using mucosal vaccines include a long-lasting immunological memory, the development of herd immunity due to secondary contact immunization, and easy administration [171].

### 3.3. Parasite Immune Modulation of the Host Mucosal Microenvironment

The gastrointestinal tract in ruminants is home to a variety of commensal microorganisms that play a large role in host nutrition, homeostasis, and the development of the host’s immune system [172]. The effect of GIN infections in the gut can be quantified in the increase or decrease of alpha and beta diversity in the host gut that signifies the regular mean microbial diversity and the ratio between the normal and nematode-infected microbial species diversity, respectively [172,173]. It has been shown in several GIN infections that infecting nematodes are able to modulate the population of the host microbiome to their advantage. In a trial vaccination study using *T. circumcincta* in lambs immunised with a cocktail of recombinant antigens, 16s rRNA sequencing of faecal samples from both immunised and control groups was carried out [161,173]. Results showed an overrepresentation of *Prevotella* spp. in vaccinated and adjuvant groups compared to untreated groups post-challenge [173]. Members of the genus *Prevotella* are associated with peptide degradation, and their increase is hypothesised to compensate for low protein levels during *T. circumcincta* infections [172]. It has been proposed that communication between infecting nematodes and gut microflora is facilitated by helminth derived extracellular vesicles (EVs) that contain immunomodulatory factors, such as transforming growth factor β (TGF-β) and peroxiredoxins, that induce responses to infection and regulate anthelmintic type-2 responses [33]. EVs have also been implicated in parasite migration during changing larval stages and for nutrition, and their presence in the gut can affect a number of beneficial commensals such as probiotics [174]. A deeper understanding of how parasite mucosal vaccines are affected by changes to the microbiome could help develop new vaccine antigens and increase the efficiency of vaccine delivery in the host.

## 4. Conclusions

The economic losses associated with GIN infections are innumerable and adversely affect smallholder livestock farmers who rely on sheep and goats for daily subsistence. Due to their complex biology and the haphazard use of anthelmintic drugs that has become prevalent in recent years, there has been an increase in the spread of drug resistant nematode infections globally. New effective integrated control measures are necessary to curb the spread of drug-resistant GIN infections. Vaccination combined with pasture management presents a viable option, with whole parasite or native protein antigens showing effective control of circulating GIN infections. Successful recombinant vaccines have yet to be realised due to the poor understating of prokaryotic post-translational modifications and host–parasite interactions that GIN antigens require due to the complex biology of nematodes. Whole parasite irradiated vaccines delivered through the mucosal route present a quicker and already proven way to start rolling out effective vaccines where they are most required, as recombinant versions are still under development. For mucosal vaccines, irradiated non-dividing but metabolically active L3 stage larval vaccines can currently be used without adjuvant for oral vaccination. Purified antigens can also be used orally with adjuvant to protect them from degradation in the host gut or intravenously to develop protective anti-parasite antibodies that can access the parasite gut during a blood feed. Targeting the mucosal immune system would be of extreme importance in controlling the GI parasitic infections in small ruminants since there is a potent immune induction, especially by native antigens. Cells such as mucosal epithelial cells, intraepithelial leucocytes, submucosal mast cells, submucosal eosinophils, B cells and T cells in MALT and in local lymph nodes contribute to ensuring a quick and effective immune response. In addition, chemical mediators such as mucins, cytokines and antibodies also play a role in both innate and adaptive responses. Combining these proven vaccine prototypes with pasture management will buy farmers more time and help control GIN infections where they are most prevalent.

There are, however, several limitations to the proposal suggested here. Irradiation studies for many of the GIN parasites were carried out using basic irradiation parameters that could result in mostly dead parasites used for immunisation. Current irradiation methods can be fine-tuned to now produce metabolically active but non-dividing parasites at much lower doses than previously done [117]. Many of the previously attempted parasite experiments carried out between the 1960s and the 1980s would benefit from a second attempt due to improved technology [175]. A second limitation of this work is the lack of in-depth assessment of immune responses in protected test subjects immunised with irradiated whole parasites. Insights into the protective mechanisms involved would educate future researchers working on recombinant vaccines on which responses are important to target for protection. A third limitation arises from the fact that many experiments were carried out either in surrogate test animals, such as mice and guinea pigs, or only in sheep and not in goats that are infected by a different mix of GIN infections [9]. In addition, goats are browsers and rely on different plant sources for nutrition whereas sheep are grazers that mostly rely on grass therefore making the microenvironment and consequent host immune responses different for the same GIN infection [176]. New studies would be required to close these gaps in knowledge for whole parasite and native antigens using several new techniques such as the development of Ovine organoids to study host-parasite interactions in vitro [177].

## Figures and Tables

**Table 1 ijms-25-01409-t001:** Irradiated nematode vaccines.

Nematode (L3 Larvae)	Host	Dose	Outcome (Ref)
*Ancylostoma caninum*	Dog	1 × 10^3^ at 400 Gy (X-ray and ^60^C)	90% protection after 2 vaccinations. Discontinued due to poor commercialisation [120,121,122,123,124,130]
*Haemonchus contortus*	Sheep	1 × 10^4^ at 300–600 Gy (^60^Co)	86% in animals older than 6 months after 2 vaccinations, poor responses in 2-month-old lambs [125,126,127,130]
*Trichinella spiralis*	Pig	5 × 10^4^ at 100 Gy (^60^Co)	Protection from challenge [130,131]
*Trichostrongylus colubriformis*	Sheep	2 × 10^4^ at 600 Gy	78% level of protection in lambs challenged 1 month after 2 vaccinations [129]
*Nematodirus battus*	Sheep (6 weeks old)	2 × 10^4^ at 600 Gy	66% reduction in worm burden when challenged with 50,000 infective larvae 1 month after vaccination [132]
*Strongylus vulgaris*	Horse	70–100 Gy	Recovery of worms was lower in vaccinated ponies and clinical sings were less obvious [133,134]
*Dictyocaulus viviparus*	Cattle	1 × 10^3^ at 400 Gy	Reduced worm burdens by 95% for 12 months, subsequent natural infections led to life-long immunity and is successfully commercialised [116,135,136]
*Dictyocaulus filaria*	Sheep	1 × 10^3^ at 400–500 Gy	2 vaccinations 1 month apart showed a high degree of resistance against *D. filaria* and were also protected to some degree against infection with other nematodes. Commercially available as Bovilis^®^ [137,138]
*Bunostomum trigonocephalum*	Sheep	6 × 10^3^ 200–600 Gy (^60^Co)	Increasing doses of irradiation had a corresponding decrease in the worm burden [139]
*Oesophagostomum columbianum*	Sheep	5 × 10^2^–2 × 10^3^ 400 Gy (^60^Co)	72% protection after 2 vaccinations 21 days apart [140,141]

**Table 2 ijms-25-01409-t002:** Parasite protein products used as vaccine targets.

Nematode Protein	Host	Dose	Outcome (Ref)
*H. contortus* H11/H-gal-GP parasite gut proteins	Sheep, goats, cattle, alpacas	5 µg × 6 vaccinations	Requires improved nutrition for pregnant and lactating ewes; effective results of up to 80% reduction in EPG; commercially available as Baebervax^®^ [13,142,143,144,145,146,147,148,149,150]
*H. contortus* Combined; Hco-gal-m Hco-gal-f	Sheep, goats, cattle, alpacas	5000 L3	41–46% worm burden reduction, 37–48% faecal egg count reduction [157]
*H. contortus* Hc23	Sheep, goats, cattle, alpacas	15,000 L3	70.67–85.64% faecal egg count reduction, 67.1% and 86% worm burden reduction [158]
*Teladorsagia circumcincta*; mixed larval stage metabolites and ES from in vitro culture	Sheep	300,000 Larvae	44% worm burden reduction, 81% faecal egg count reduction [159]
*Teladorsagia circumcinctal*; L3 soluble gut membrane proteins Oc-gal-GP	Sheep	140 µg	8% worm burden reduction, 28% faecal egg count reduction, H contortus antigens cross protection [22,160].
*Teladorsagia circumcincta*; 8 recombinant antigen cocktail: Tci-APY-1, Tci-ASP-1, Tci-CF-1, Tci-ES20	Sheep	50 µg each protein	55% worm burden reduction, 70% faecal egg count reduction [22,161]
*T. colubriformis* Tc L3 Homogenate	Sheep	30,000–50,000 larvae	Cellulose adjuvant—30% worm burden reduction, 49–53% faecal egg count reduction; chitosan adjuvant—10% worm burden reduction, 10–25% faecal egg count reduction [162,163]
*T. colubriformis* Adult ES antigens	Sheep	100 µg	47% worm burden reduction, 52% faecal egg count reduction [162,163]
*Oesophagostomum radiatum* Live in-vitro-grown L3 or L3–4 mixtures	Cattle	50 µg × 3 vaccinations	L3—44 to 90% faecal egg count reduction; L3–4 mixtures—36 to 83% faecal egg count reduction [164]
*O. ostertagi* Polyprotein allergen(OAP)	Cattle	100 µg × 3 vaccinations	60% reduction of egg count for at least 2 months [151]

## Data Availability

No new data were created or analysed in this study.

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
