# Peer review of "Mucosal Immunity of Major Gastrointestinal Nematode Infections in Small Ruminants Can Be Harnessed to Develop New Prevention Strategies"

_ijms, 2024, doi:10.3390/ijms25031409_

Round 1

Reviewer 1 Report

Comments and Suggestions for Authors

Reviewer 2 Report

Comments and Suggestions for Authors

Dear authors,

The manuscript title is “MUCOSAL IMMUNITY OF MAJOR GASTROINTESTINAL NEMATODE INFECTIONS INSMALL RUMINANTS CAN BE HARNESSED TO DEVELOP NEW PREVENTION STRATEGIES and it aims to review the most prevalent GINs in small ruminants as well as the different prevention strategies that exist.

The topic falls within the aims and scope of the journal and it is important in the context of anthelmintic resistance. The abstract is clear and complete. The manuscript misses a very important part: Methodology. The authors must explain how did they select the papers to do this review. Maybe it can explain the absence of recent important papers on this matter – from Europe, it is right, but should be stated here. Also, some of the references used are very very old and not so credible. The authors need to substitute or reduce the use of the reference 25.

Some particular suggestions/comments will be done here:

- Line 23 – I suggest the authors not to repeat in keywords words that are already in the title

- Lines 26/ 27 – Please delete the repeated words

- Line 29 – Please use GIN as you have already written before, in this line and from here on

- Lines 31/32/85 and others, please review all – Please pay attention to taxonomy. sp and spp are not in italic

- Line 38 – infestation is not correct, correct with infection please

- Line 72/73 – the authors may say here that these are two species found in the intestine

- Line 81 – “and”, no italics

- Lines 88/ 89 and before – you refer always the same symptoms, maybe you can only refer the symptoms are the same/similar for this group of parasites

- Lines 97 / 107 – 10 lines with the same reference is not correct, especially in a review paper

- Lines 109 / 111 – this is concerning which GIN?

- Lines 111/112 – your title says your manuscript is about small ruminants. You should focus on small ruminants not in cattle and other, as you previously had stated, there are already sufficient and unexplored differences between sheep and goats and those should be presented here.

- Lines 112/114 – this is against all the studies this reviewer knows. Looking at the reference, beside being from 1995, it is from a congress, not the best option to a review article when there are thousands of excellent and credible papers about it

- Line 119 – doses of parasites is not correct

- Line 138 – Trichinella (in ruminants??) is not a GIN

- Line 241 – please write first ES in full

- Line 396 – please write the genus in full as H. maybe also Haemonchus

- Line 425 – you stated only three

- Lines 461 /480 – again 20 lines with the same reference in a review manuscript is not correct

- Lines 508/509 – gerbils are not small ruminants…

- Line 511 – use only ES

- Line 568 – add italics

- From Line 602 – on: some references are too old, we can not approach anthelminthic resistance and vaccines or epidemiology with studies more than 20 years old. References nr. 35/43 are from 1985/1984! Imagine how much immunology developed since then! Some are not complete and we can not understand what they are (example: nr. 6, 144)

Round 2

Reviewer 2 Report

Comments and Suggestions for Authors

Dear authors,

Thank you for your review.

No changes have been made in this reviewer major concern:

The manuscript misses a very important part: Methodology. The authors must explain how did they select the papers to do this review.

If this is a review article the authors must explain how did they did it. Is it a systematic revision? Is it not? Inclusion and exclusion factors, among others.

Author Response

Dear Reviewer 2,

We would like to confirm that this review is not systematic but rather exploratory as several recent reviews have thoroughly explored the subject of GIN infections. We wanted this work to make the connection between proven vaccination studies that already work quite well but have not been explored with most current vaccination trials looking at recombinant protein solutions. However, there is an immediate need to jumpstart prevention even as we wait for future successes due to worsening issues with drug resistance. We have therefore added the following sentences to lines 89-95 of the resubmitted draft; "This review is narrative and discusses publications on the most common GIN parasites with an economic impact on small ruminants. We describe immune responses in the natural host or in experimental surrogates during infection with an emphasis on mucosal immunity. In addition, we cite examples of proven studies on irradiated whole parasites and purified native proteins for vaccination as part of a potential integrated parasite control strategy currently required in regions with increasing parasite drug resistance. This would serve as a solution before more effective methods of control can be discovered. "

I hope this addresses the major concern on methodology.

Round 3

Reviewer 2 Report

Comments and Suggestions for Authors

Dear authors,

Narrative reviews have several limitations, if you have decided by it, you should then state these limitations in discussion/conclusions.

Best regards

Author Response

Dear Reviewer 2,

the following text was added to lines 633-651; 

"There are, however, several limitations to the proposal suggested here. Irradiation studies for many of the GIN parasites were carried out using basic irradiation parameters that could result in mostly dead parasites used for immunisation. Current irradiation methods can be fine-tuned to now produce metabolically active but non-dividing parasites at much lower doses than previously done [117]. Many of the previously attempted parasite experiments carried out between the 1960s to the 80s era would benefit from a second attempt due to improved technology [175]. A second limitation of this work is the lack of in-depth assessment of the immune responses in protected test subjects immunised with irradiated whole parasites. Insights into the protective mechanisms involved would educate future researchers working on recombinant vaccines on which responses are important to target for protection. A third limitation arises from the fact that many experiments were carried out either in surrogate test animals, such as mice and guinea pigs, or only in sheep and not in goats that are infected by a different mix of GIN infections [9]. In addition, goats are browsers and rely on different plant sources for nutrition whereas sheep are grazers that mostly rely on grass therefore making the microenvironment and consequent host immune responses different for the same GIN infection [176]. New studies would be required to close these gaps in knowledge for whole parasite and native antigens using several new techniques such as the development of Ovine organoids to study host-parasite interactions in vitro [177]."